# Unveiling the immunomodulatory properties of starch microparticles on alveolar macrophages

Alejandra Barrera-Rosales[1,2], Dulce Mata-Espinosa[3], Vanessa Villegas-Ruiz[1], Mayra Silva-Miranda[4,5], Edgar Zenteno[6], Sergio Sánchez[1], Rogelio Hernández-Pando[3], Romina Rodríguez-Sanoja[1*], Silvia Moreno-Mendieta[1,5*]

**1** Departamento de Biología Molecular y Biotecnología, Instituto de Investigaciones Biomédicas, Universidad Nacional Autónoma de México (UNAM), Ciudad Universitaria, Ciudad de México, México, **2** Doctorado en Ciencias Bioquímicas, Universidad Nacional Autónoma de México (UNAM), Ciudad Universitaria, Ciudad de México, México, **3** Sección de Patología Experimental, Instituto Nacional de Ciencias Médicas y Nutrición Salvador Zubirán, Delegación Tlalpan, Ciudad de México, México, **4** Departamento de Inmunología, Instituto de Investigaciones Biomédicas, Universidad Nacional Autónoma de México (UNAM), Ciudad Universitaria, Ciudad de México, México, **5** Secretaría de Ciencia, Humanidades, Tecnología e Innovación (SECIHTI), Colonia Crédito Constructor, Ciudad de México, México, **6** Departamento de Bioquímica, Facultad de Medicina, Universidad Nacional Autónoma de México (UNAM), Ciudad Universitaria, Ciudad de México, México

\* moreno.sa@iibiomedicas.unam.mx (SMM); romina@iibiomedicas.unam.mx (RRS)

## Abstract

Polysaccharides as immunomodulators are increasingly explored in preclinical studies, showing potential applications for preventing or treating different diseases. Among them is starch, an α-glucan formed by amylose and amylopectin chains. Given their abundance in nature, physicochemical characteristics, and applicability in pharmacy, they are versatile molecules that offer important biotechnological and biomedical advantages. Most studies about starch immunostimulant properties focus on modified-soluble and particulate α-glucans. However, little research has been done on the immunostimulant properties of starch in its natural particulate state. Previously, we have used starch microparticles (SMPs) as carriers for nasal administration of antigens in healthy mice and as a nasal boost and adjuvant of the Bacillus Calmette–Guérin (BCG) vaccine in a murine model of tuberculosis. This study aimed to analyze the effect of SMPs on the activation and polarization profile of murine alveolar macrophages of the MH-S cell line. We evaluated the effect of these SMPs on cell viability, phagocytosis, and expression of surface markers on M0 alveolar macrophages. We also explored the effect of SMPs on nitric oxide production, cytokine secretion, glucose consumption, and lactate release on M0 and previously M1 and M2-polarized alveolar macrophages. The results indicate that these SMPs are phagocytosed without cytotoxic effects for alveolar macrophages and have an immunomodulatory effect on previously polarized M1 macrophages. In M0 and M2 macrophages, the SMPs induced a mixed secretion of cytokines such as TNF-α, IL-10, and IL-12p40, and a significant decrease of TGF-β1. Recognizing the effects triggered

**Data availability statement:** All relevant data are within the paper and its Supporting Information files. The Supporting Information files contain the raw data behind the means and statistical data used to build graphs and replicate the results of the study.

**Funding:** SMM was supported by grant A1-S-14446 Ciencia Básica (SEP-CONAHCyT), RRS was supported by grants A1-S-9849 Ciencia Básica (SEP-CONAHCyT) and IN 216722 PAPIIIT/DGAPA/UNAM. ABR was supported by doctoral scholarship 894774 (CONAHCyT). There was no additional external funding received for this study.

**Competing interests:** The authors have declared that no competing interests exist.

by these SMPs on these cells of the innate immune system will allow us to propose rational uses for these SMPs in prophylactic and therapeutic vaccines intended to be used by the nasal/pulmonary route.

## Introduction

In recent years, the study of the properties and advantages of using polysaccharides as immunomodulators has increased. In general, these molecules possess desirable biocompatibility, are well tolerated, can be manipulated in their solubility characteristics, and some of them are mucoadhesive. All these advantageous properties offer the opportunity to develop carrier systems for systemic or mucosal administration [1]. Some polysaccharides showing immunostimulatory activity include chitosan [2], inulin [3], mannans [4], galactans [5], fructans [6], and α and β-glucans [7,8]. A remarkable example is delta-inulin, which showed a good safety profile and efficacy in preclinical studies [9] and has been approved as an adjuvant for the first protein-based COVID-19 vaccine (SpikoGen®) [10].

For glucans, the observation that the β-1,3 glycosidic linkage is not an exclusive requirement for the immunostimulatory activity and that α-glucans may interact with the same receptors on professional antigen-presenting cells (APCs) that β-glucans do [11,12], has motivated the research in the field to determine the effect of chemically modified α-glucans on cellular activation, proliferation, increased endocytosis and production of pro- and anti-inflammatory cytokines and antimicrobial molecules [13,14]. Those effects on cells help to explain the adjuvant or immunomodulatory properties reported for maltooligosaccharides [15,16], soluble glucans purified from plants and fungi [17–19] and based-starch nano and microparticles [20,21], which clearly shows their potential for the formulation of adjuvants or therapeutics [22–24].

In our previous work, we used non-functionalized starch microparticles (SMPs) as antigen vehicles for oral and nasal administration. The antigens we used are non-covalently immobilized on these SMPs and allow the induction of antigen-specific immune responses after administration in healthy mice [25,26]. We also used these SMPs as a nasal boost of the BCG vaccine in *Mycobacterium tuberculosis*-infected mice, achieving a decrease in the bacillary load and pneumonia in the lungs of boosted mice [27], strongly suggesting that SMPs without immobilized antigen enhanced the protective efficacy of the BCG vaccine. These findings raise important questions and encourage us to uncover the mechanisms behind these *in vivo* responses. Recently, it has been shown that these SMPs cross the epithelial cell barrier and reach the Mucosal Associated Lymphoid Tissue (MALT) after oral and nasal administration [28], which implies that they can interact with the immune cells present at the nasal- and gut-associated lymphoid tissue. One of the main cell types there are macrophages, which are important innate immune cells with versatile functions and plasticity that change from one phenotype to another depending on the microenvironment to maintain homeostasis. In this work, we aimed to evaluate the effect of SMPs on murine alveolar macrophages of the MH-S cell line activation and phenotype modulation. This is very important to open the usage perspectives and opportunities for research and design of safe adjuvants and targeted treatments with SMPs.

## Materials and methods

### Cell culture, maintenance, and polarization

Murine alveolar macrophages line MH-S (ATCC CRL®-2019TM) were cultured in 75 cm$^2$ treated flasks with 12 ml RPMI 1640 medium (Gibco, 31800−014) supplemented with 10% (v/v) heat-inactivated fetal bovine serum (FBS) (ByProducts Lot 21001) and 1% Anti-Anti 100x (Capricorn, CP22−5084) and maintained at 37°C, 95% relative humidity and 5% $CO_2$. For each assay, the cells were gently detached with a scraper (Corning Incorporated Costar, 3010) with fresh complete RPMI medium, and cell counting was performed with 0.4% trypan blue using a Neubauer chamber. Before polarization, macrophages were cultured for 6 h for adherence at 37°C, 95% relative humidity, and 5% $CO_2$. Polarization towards M1 phenotype was induced 18 h in a complete RPMI medium supplemented with 100 ng/ml LPS (SIGMA, *E. coli* strain O111:B1, L4391-1MG) and 20 ng/ml IFN-γ (Invitrogen, Thermo Fisher Scientific, BMS326). For polarization towards the M2 phenotype, macrophages were incubated for 18 h in complete RPMI medium supplemented with 20 ng/ml IL-4 (Sigma-Aldrich, I1020) and 20 ng/ml IL-13 (Stemcell, 78030.1).

### Starch microparticles (SMPs) preparation

Ten mg of SMPs (SIGMA-ALDRICH S7260-500G) were weighed in Eppendorf tubes and dry sterilized. They were washed three times with Milli-Q® water, once with RPMI, and resuspended in 1 ml of RPMI. Stocks of 1 mg/ml and 100 μg/ml were prepared. Counting was performed for the 100 μg/ml stocks in the Neubauer chamber to estimate the number per stock. The ratio of three SMPs per cell (1:3) was used for all the experiments.

### Cell viability assay

We evaluated the impact of the addition of SMPs on MH-S macrophage viability. For this, we added increasing doses of SMPs in M0 MH-S cells. Briefly, in a 96-well plate, $3 \times 10^4$ cells per well were seeded and left overnight at 37°C, 95% relative humidity, and 5% $CO_2$. Cells of three wells were detached and counted to calculate cell-to-SMP ratios of 1:1, 1:3, 1:5, 1:10, 1:20, and 1:50. The antibiotic puromycin (500 μg/ml) was used as a toxicity control. We followed the MTT protocol according to the manufacturer's instructions (V13154, Invitrogen, Thermo Fisher). Briefly, after 24-hour incubation with the stimuli, we incubated all the groups with MTT reagent for 4 h. The crystals of formazan were dissolved with SDS and the concentration was determined by optical density at 570 nm. Culture medium and culture medium with SMPs alone were used for background subtraction in all groups. Cell viability was calculated as % cell viability = [((Absorbance of sample – Absorbance of RPMI+alone SMPs) x 100)/Absorbance of control].

### Phagocytosis assay

The phagocytosis of SMPs by MH-S was evaluated by flow cytometry and transmission electron microscopy (TEM). For flow cytometry analysis, the SMPs were FITC-labeled (FITC-SMPs), as previously described [28]. Briefly, 10 mg of sterilized SMPs were washed 3 times with Milli-Q® water and suspended in 1 ml of 0.1 M sodium carbonate buffer pH 9 with 1 mg/ml FITC (previously dissolved in DMSO) incubated overnight at 4°C with orbital shaking and protected from light. The reaction was stopped with 50 mM $NH_4Cl$ for 2 h at 4°C, protected from light. SMPs were washed 6 times with sterile Milli-Q® water and counted in a 100 μg/ml stock to perform the corresponding calculations for stimuli. In a 12-well plate, $5 \times 10^4$ cells per well were seeded and left overnight at 37°C, 95% relative humidity, and 5% $CO_2$ for adherence and growth. As inhibition control for phagocytosis, before stimulation with SMPs, the control groups were preincubated with 20 μM of cytochalasin D (CytD) for 1 h under the same conditions. After this time, FITC-SMPs were added in RPMI without FBS in a 1:3 cell: SMPs ratio, and phagocytosis was evaluated at 3 and 24 h. In the first case (3 h), the cells were detached by gentle pipetting with RPMI, washed once with sterile PBS 1X and stained; in the second case (24 h), after 3 h of incubation with SMPs, the cells were washed once with PBS 1X and fresh complete RPMI medium was added until

24 h of incubation. After this time cells were also detached by gentle pipetting with RPMI, washed once with sterile PBS 1X and stained. Viability staining was performed with 0.05 µg/ml FVS 575V (565694, BD Horizon™) for 30 min at room temperature (RT) protected from light. One wash with FACs buffer (PBS 1X, 2% SFB, 0.1% NaN3) was done and then the cells were stained with APC-F4/80 for 30 min at 4°C protected from light. After this time, cells were washed once with FACs buffer, and the samples were fixed in 1% paraformaldehyde (PFA). All the experimental groups were stained with viability FVS 575V and APC-F4/80, phagocytosis events were considered only those with double staining APC-F4/80+ and FITC-SMPs +. The reading was performed in a flow cytometer Attune NXT (LabNalCit, UNAM). Data was analyzed with FlowJoTM Software, version 10 (Becton Dickinson, Ashland, OR, USA).

For TEM analysis of phagocytosis, $1.5 \times 10^6$ cells were seeded in 25 cm$^2$ treated flasks with 3 ml of complete RPMI and incubated overnight. For analysis control groups were preincubated with 20 µM of CytD for 1 h. Then, all the groups were incubated with SMPs in a 1:3 ratio for 3 h. After incubation, the cells of groups of 3 h were gently detached, washed, and then fixed with 1 ml of cold 2.5% glutaraldehyde solution in Cacodylate buffer (0.15 M pH 7.2) for 5 min at RT. Cells of the 24 h group were washed 3 times with 2 ml of PBS 1X and fresh complete RPMI medium was added to continue incubation for 24 h. After incubation, the cells were gently detached, washed and fixed as described. Finally, cells were centrifuged for 5 min at 1500 rpm, 1 ml of the cold fresh 2.5% glutaraldehyde solution was added, and the cells were allowed to fix for 24 h at 4 °C. For ultrastructural analysis, the post-fixation was performed with 1% Osmium Tetraoxide. The samples were dehydrated with ethyl alcohol at increasing concentrations; inclusion was made in Spurr Low Viscosity Resin (Electron Microscopy Sciences, Fort Washington, PA). The ultrathin sections were made in thickness of 80 nm in copper grids that were contrasted with 1% uranyl acetate (Electron Microscopy Sciences, Fort Washington, PA) and with Lead Citrate "Reynolds" for examination with a JEOL JEM 1200EX II electron microscope. Quantification of SMPs phagocytosed by macrophages was performed in semithin sections (0.5–1 µm) stained with toluidine blue under a light microscope, counting macrophages with and without SMPs present in 100-cell fields in triplicate.

## Analysis of macrophage immunophenotype

For analysis of expression markers, $5 \times 10^4$ MH-S macrophages per well were seeded in a 12-well plate in 1 ml of complete RPMI medium and incubated overnight at 37°C, 95% relative humidity and 5% $CO_2$. Cells were stimulated with SMPs in a 1:3 ratio for 24 h. After incubation, cells were detached by gentle pipetting with RPMI, washed once with sterile PBS 1X, and stained. The cell suspensions were incubated with the viability stain FVS 575V (565694, BD Horizon™) for 30 min at RT protected from light and then with the antibody cocktail for 30 min on ice, protected from light. M0 markers were identified with specific anti-mouse antibodies for APC-F4/80 (Invitrogen Thermo Fisher, 17-4801-82) and Alexa-Fluor 700-CD11c (BD Pharmigen, 560583). M1 macrophage phenotype was identified with specific anti-mouse antibodies Pe-Cy5-CD80, APC-eFluor780-CD86, FITC-MHC-II and PerCP-eFluor 780-CD369 (Invitrogen Thermo Fisher, 15-0801-82, 47-0862-82, 11-5322-82, 46-5859-82, respectively) and M2 macrophage phenotype was detected with eFluor 450-CD206 and PE-CD209 (Invitrogen Thermo Fisher, 67-1631-82, 48-2061-82, 12-2092-82). Concentrations were standardized for each antibody. Cells were washed with FACS buffer, and the samples were fixed in 1% PFA for reading using the flow cytometer Attune NXT (LabNalCit, UNAM). Data were analyzed with FlowJoTM Software, version 10 (Becton Dickinson, Ashland, OR, USA).

## Quantification of Nitric Oxide

For nitric oxide (NO) determination in supernatants, $5 \times 10^4$ cells per well were seeded in a 12-well plate in 1 ml of complete RPMI medium and incubated for 6 h for adherence. Cells were incubated with stimuli for 18 h for polarization M1/M2 at 37°C, 95% relative humidity and 5% $CO_2$. The M0, M1, and M2 cells were stimulated with SMPs in a 1:3 ratio, and 250 µl of supernatants were collected at 3, 24, and 48 h. Two control groups were stimulated with 10 µg/ml of soluble starch

(SS) as α-glucan and 10 µg/ml of zymosan (Zym) as β-glucan structures. After incubation, supernatants were collected and filtered by 0.22 µm polyethersulfone (PES) membranes and frozen at −70 °C until use. The NO was determined by the Griess method [29] using a sodium nitrite standard curve and the reagents sulphanilamide (SIGMA-ALDRICH, S9251-100G) and N-(1-Naphthyl) ethylenediamide dihydrochloride (SIGMA-ALDRICH, 222488-5G). Experiments were performed in triplicate.

### Glucose consumption and lactate release

Culture and polarization conditions were performed as described before. The M0, M1 and M2 cells were stimulated with SMPs in a 1:3 ratio, 10 µg/ml SS and 10 µg/ml Zym. Fresh supernatants were collected from the same wells at all the times of incubation (3, 24, and 48 h) and immediately used for the enzymatic colorimetric methods to determine glucose using the Glicemia Enzimática AA Kit (140060, Wiener Laboratories) and lactate release using the Lactate Kit (1999795, Wiener Laboratories) according to the manufacturer's instructions. The consumption of glucose was determined as the reduction of glucose levels in supernatants in comparison with RPMI 10% FBS medium and was calculated regarding the glucose diminishing in control macrophages incubated only with complete RPMI medium. Experiments were performed in triplicate.

### Quantification of cytokines

The same supernatants for NO quantification were used to determine cytokines using a flow cytometry-based analysis (Luminex®). The Milliplex® Mouse Cytokine/Chemokine Magnetic Bead Panel (MCYTOMAG-70K) for IL-1β, IL12p40, IL-10, and TNF-α, and the Milliplex® TGF-β1 Single Plex Magnetic Bead Kit (TGFBMAG-64k-01) for TGF-β. Blanks, standards, quality control, and samples were processed following the manufacturer's instructions (Merck). The xPonent® MAGPIX software was used for the analysis of the Logistic 5P Weighted standard curves and the samples. As indicated before, the experiments were performed in triplicate.

### Statistical analysis

Data were presented as means ± standard error of the mean (SEM) and included triplicate experiments. Statistical analysis was performed with GraphPad 8.0.1 Software. For most experiments, a Two-Way Analysis of Variance followed by a Multiple Comparison test was carried out. Details for statistical analysis are given in each figure. P-values <0.05 were considered statistically significant. See S2 File for the raw data underlying Figs 1–7.

## Results and discussion

### The addition of SMPs does not affect the cell viability of MH-S alveolar macrophages

In Fig 1, we show the impact of SMPs on the viability of MH-S macrophages. These macrophages exhibited high viability rates ranging from 91% to 100% following the addition of SMPs, irrespective of the increasing concentrations. While it is unlikely that alveolar macrophages *in vivo* would encounter large amounts of these SMPs, it was essential to determine the dosage that could potentially influence their viability. This assessment is pivotal, as determining the toxicity of inhalable agents on the respiratory system is a critical aspect of preclinical studies [30,31]. Upon microscopic examination, cells exposed to SMPs exhibited preserved morphology and characteristics. Conversely, treatment with the antibiotic puromycin (500 µg/ml) led to a 70% reduction in cell viability. For all subsequent experiments, we established a cell-to-SMP ratio of 1:3, which proved non-toxic to the cells. Furthermore, we considered several factors, including the average size of mouse alveolar macrophages (approximately 12 µm) and the literature indicating that particles within the 1–3 µm size are those that preferentially reach the alveoli [32]. Our results are consistent with other studies documenting the low toxicity levels of polysaccharide particles designed as delivery systems targeting macrophages [33].

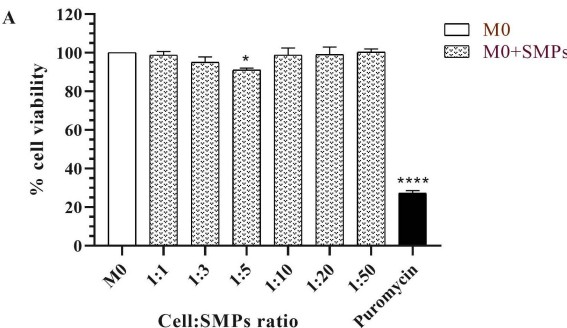

**Fig 1. Cell viability of alveolar macrophages of MH-S cell line stimulated with SMPs for 24 h**. All results are expressed as means of three independent experiments ± SEM. Viability of M0 macrophages without SMPs (smooth bar) compared with M0 macrophages added with SMPs (dotted bars). Two-way ANOVA with Dunnett's multiple comparisons (*p < 0.05, ****p < 0.0001).

## SMPs are internalized in MH-S alveolar macrophages by phagocytosis

We investigated whether the MH-S cell line phagocytosed SMPs, understanding that phagocytosis is a critical function of alveolar macrophages in clearing inhaled particulate matter and plays a vital role in their immunological activation [34]. We evaluated SMPs' phagocytosis using flow cytometry and transmission electron microscopy (TEM) at 3 and 24 h. As phagocytosis inhibition controls, we used macrophages pretreated for 1 h with 20 μM of cytochalasin D (CytD). As illustrated in Fig 2, after 3 h of SMPs stimulation, the percentage of phagocytosis of FITC-labeled SMPs was comparable between the two groups, with measurements of 13.4% for the CytD-treated group and 15.7% for the untreated group. Following 24 h of SMPs treatment, the phagocytosis percentages increased to 19.1% and 22.1%, respectively, with no significant differences observed between the groups. Based on these results, we employed a high-resolution microscopic method to confirm the inhibitory effect of CytD on actin polymerization and to observe and quantify the SMPs within the cells. For this purpose, we prepared ultra-thin and semi-thin sections for TEM analysis.

The ultrastructural analysis revealed that macrophages treated with CytD showed one or no intracellular SMPs (Fig 3A). After 3 and 24 h of treatment, the macrophages displayed rounded surfaces and formed small pseudopodia. In contrast, macrophages treated solely with SMPs showed typical pseudopodia, indicating dynamic actin protrusions of the plasma membrane as they phagocytosed the SMPs at both the 3- and 24-hour marks (Fig 3A). Interestingly, this pseudopodia formation was observed in both the cells that phagocytized and those that did not. This observation was consistent with the percentage of SMPs phagocytosed in both treatment groups (Fig 3B). Upon counting individual instances of phagocytosis (Fig 3C), we found that the most common number of particles within macrophages was one, followed by two and three SMPs. We occasionally observed macrophages containing more particles (ranging from five to eight SMPs) (S1 Fig in S1 File). All results indicate that SMPs enter the cell through the endocytic process commonly associated with particles larger than 0.5 μm. Irrespective of the difference in the size of the SMPs, the percentage of phagocytosis in this study coincides with findings from [35] who used this same cell line and reported 10% phagocytosis of 1 μm microparticles at 2 hr. Upon counting individual instances of phagocytosis (Fig 3C), we found that the most common number of particles within macrophages was one, followed by two and three SMPs. We occasionally observed macrophages containing more particles (ranging from five to eight SMPs) (S1 Fig in S1 File). All results indicate that SMPs enter the cell through the endocytic process commonly associated with particles larger than 0.5 μm. Irrespective of the difference in the size of the SMPs, the percentage of phagocytosis in this study coincides with findings from [35] who used this same cell line and reported 10% phagocytosis of 1 μm microparticles at 2 hr.

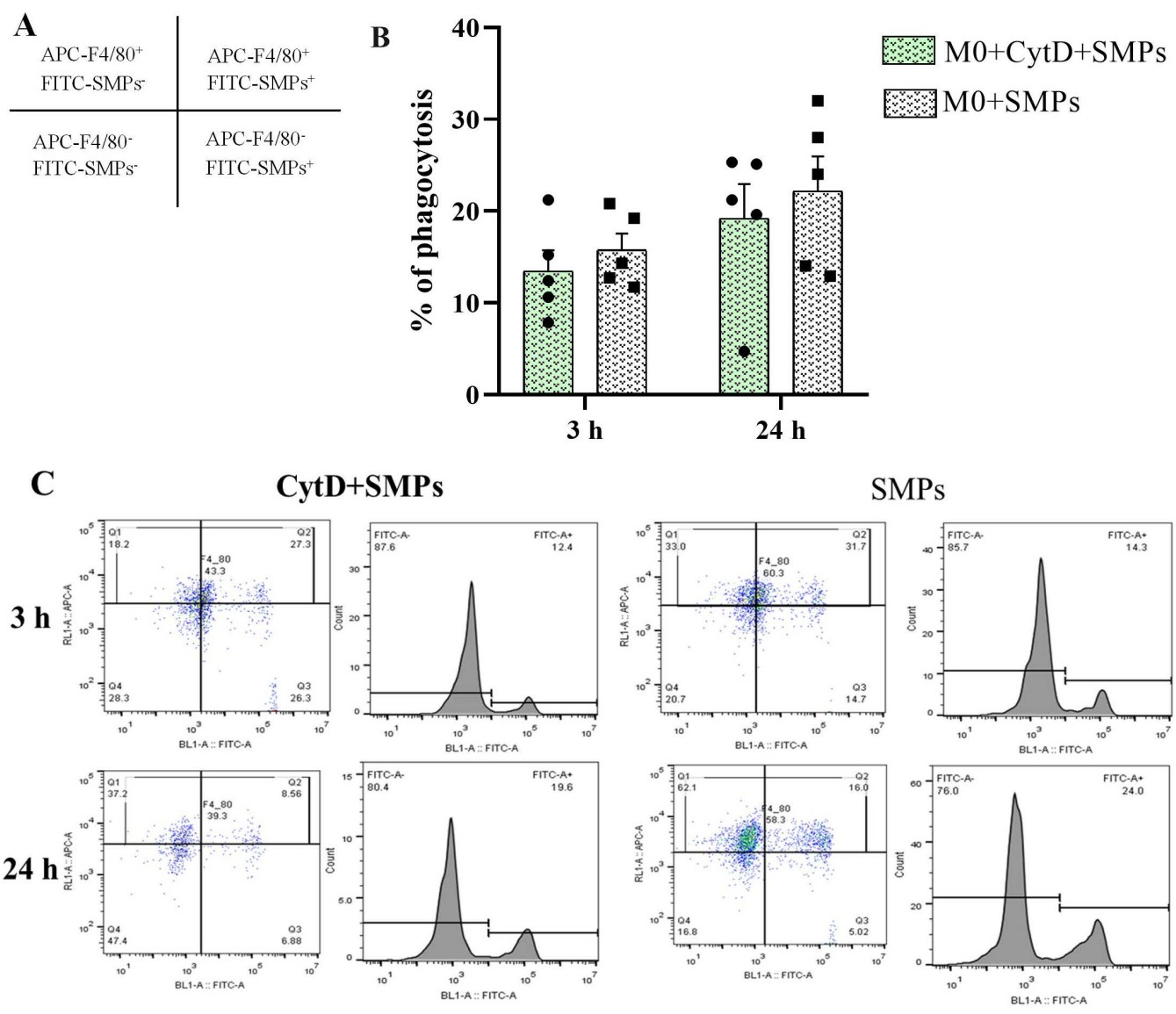

**Fig 2. Phagocytosis of SMPs by alveolar macrophages of MH-S cell line evaluated by flow cytometry. A)** Distribution of cell populations in regions based on their forward and side scatter properties and markers. The upper right quadrant shows the positive double-staining corresponding to the viable M0 macrophages (APC-F4/80⁺) and phagocytosed stained microparticles (FITC-SMPs⁺). **B)** Percentage of phagocytosis of SMPs at 3 and 24 h in M0 macrophages pretreated or not with 20 μM CytD. Bars indicate means ± SEM (n = 5). Two-way ANOVA with Sidak's multiple comparisons. **C)** Representative plots showing M0 macrophages pretreated or not with 20 μM CytD and stimulated with SMPs at 3 and 24 **h.**

Given that our SMPs were administered without prior opsonization, we ruled out an opsonic mechanism of phagocytosis. It is possible that non-opsonic receptors, such as DC-SIGN or Toll-like receptors (TLRs) like TLR2, TLR4, and TLR6, play a role. Although these TLRs are not phagocytic receptors, they may collaborate with other non-opsonic receptors to facilitate the uptake of particles; they have also been shown to recognize α-glucans [12–14,36].

Further investigation is needed to determine the exact mechanisms involved in the phagocytosis of these SMPs, which could enable the design of specific strategies to promote or inhibit phagocytosis based on their intended use.

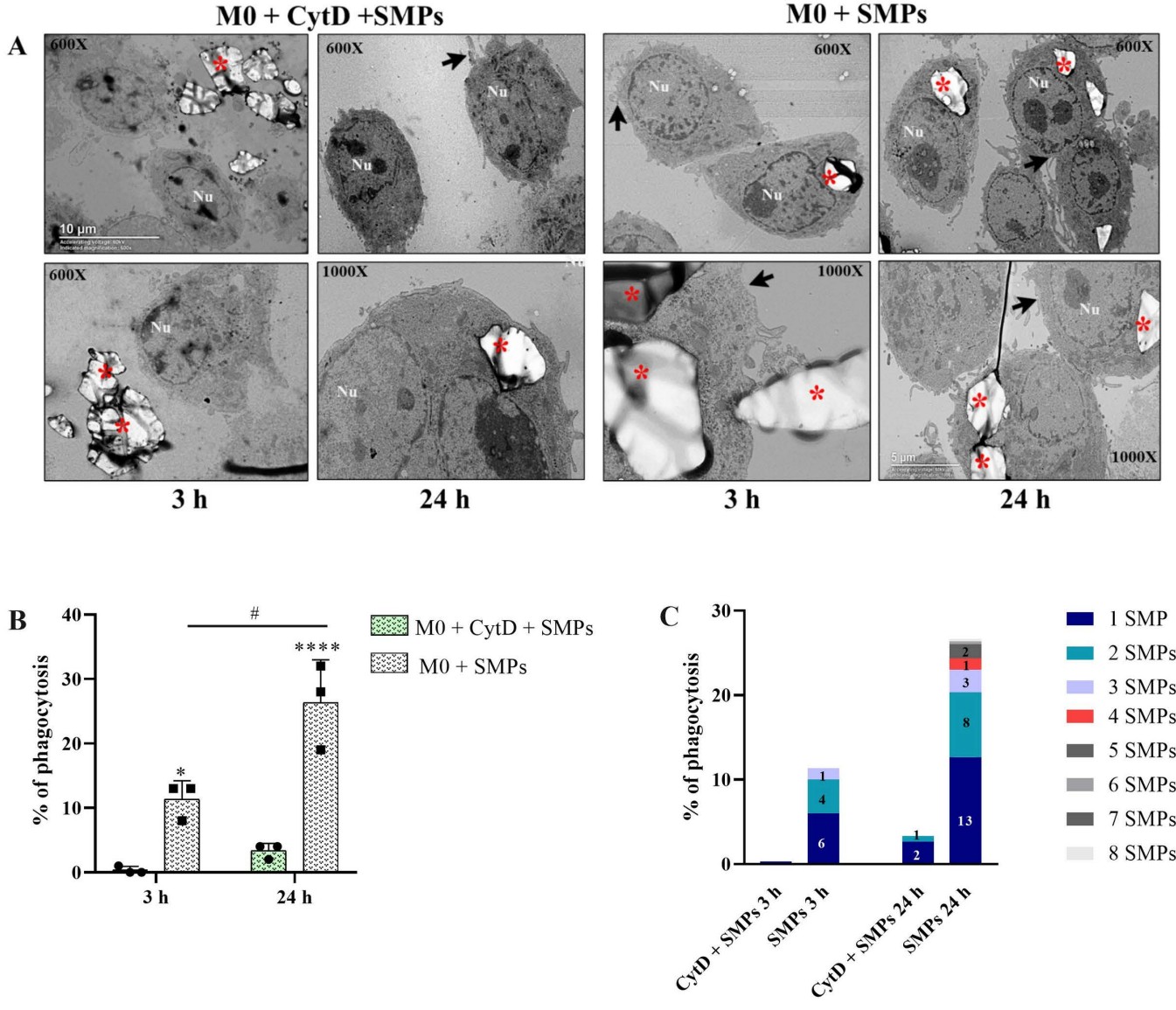

**Fig 3. Phagocytosis of SMPs by alveolar macrophages of MH-S cell line evaluated by Transmission Electron Microscopy (TEM). A)** Representative TEM micrographs showing M0 macrophages added with SMPs and pretreated or not with 20 μM CytD. Structures are indicated: SMPs (red asterisk), membrane and pseudopodia (arrowheads), and nucleus (Nu). **B)** Total phagocytosis of SMPs at 3 and 24 h in M0 macrophages compared with macrophages pretreated with 20 μM CytD. Two-way ANOVA with Sidak's multiple comparisons (*p < 0.01, ****p < 0.0001, #p < 0.05). **C)** Individual phagocytosis of SMPs. Numbers on bars indicate the number of macrophages that phagocytosed the number of SMPs indicated in colors.

## The addition of SMPs at a ratio of 1:3 does not modify the phenotype of M0 MH-S alveolar macrophages

After confirming that MH-S macrophages phagocytose the SMPs, we deemed it crucial to investigate the implications on the cells' phenotype and activation. To this end, we employed flow cytometry to assess the expression of F4/80, a well-established marker specific to murine macrophages, and CD11c, indicative of alveolar macrophages. The cell line demonstrated the expression of both markers, with F4/80 exhibiting higher levels (S2A Fig in S1 File). As illustrated in Fig 4A, this cell line also maintained expression of CD80, CD86, and MHC-II markers in a steady state (M0), and the addition of SMPs did not alter this expression.

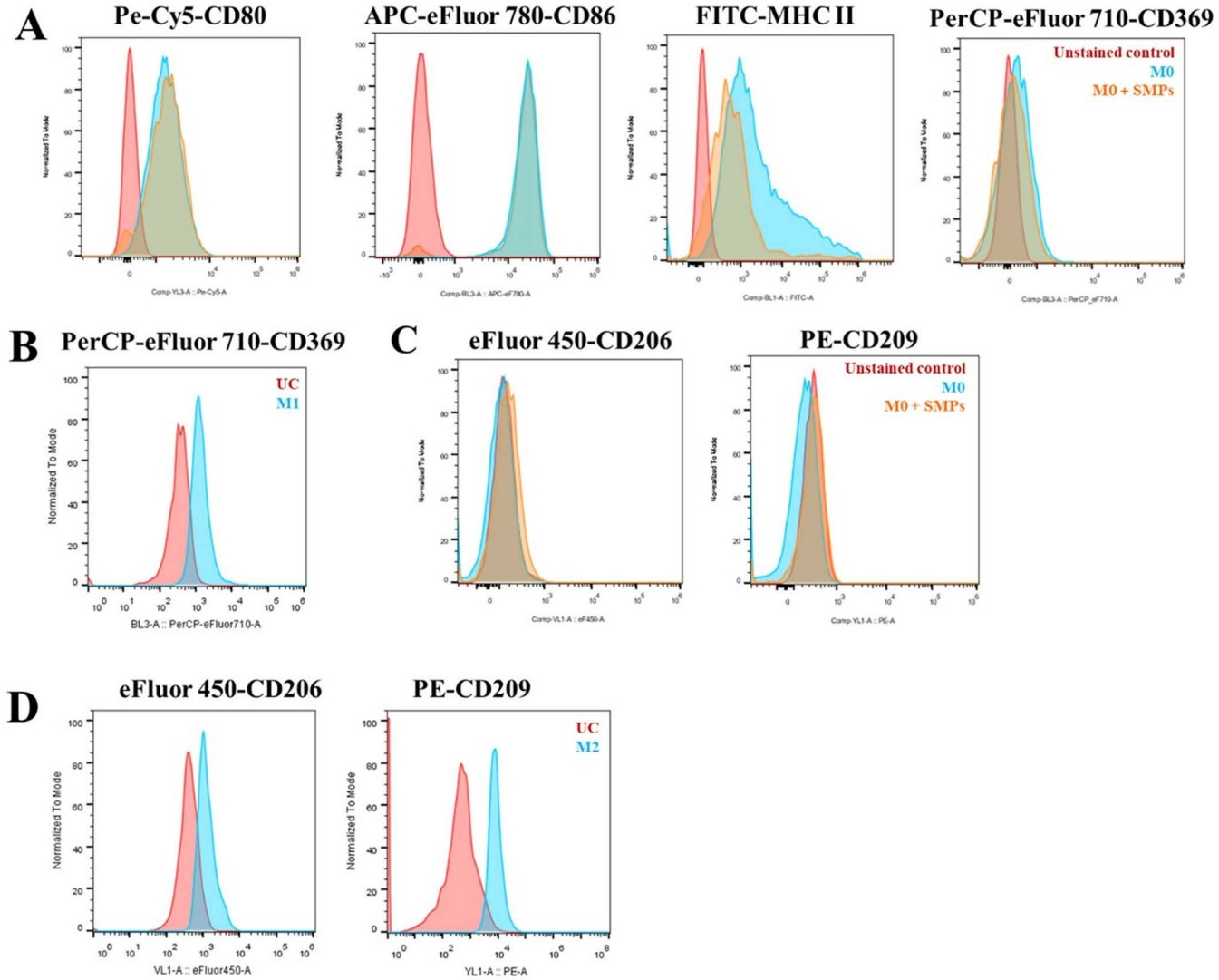

**Fig 4. Expression of M1 and M2 surface markers in alveolar macrophages of MH-S cell line stimulated with SMPs.** Representative histograms that show **A)** expression of M1 surface markers CD80, CD86, and MHC-II, and the expression of CD369 in steady macrophages with and without SMPs. **B)** expression of CD369 marker in M1 macrophages. **C)** expression of M2 surface markers CD206 and CD209 in steady macrophages with and without SMPs. **D)** expression of CD206 and CD209 in M2 macrophages.

The role of Dectin-1 (CD369) in the polarization of macrophages toward the M1 phenotype through the recognition of soluble and particulate β-glucans is well documented [37,38]. We aimed to investigate the effect of SMPs on Dectin-1 expression in M0 macrophages. Notably, this cell line does not express Dectin-1, neither at steady state nor after the addition of SMPs. To confirm this finding, we analyzed Dectin-1 expression in macrophages previously polarized to M1 phenotype with LPS and IFN-γ, where we observed significant expression (p<0.001) (see Fig 4B and S2B Fig in S1 File).

We also examined the expression of M2 markers in both M0 macrophages—those treated with SMPs—and macrophages that had been polarized to M2 using IL-4 and IL-13. The MH-S cell line does not express the mannose receptor (CD206) or DC-SIGN (CD209) markers under steady-state conditions, and this lack of expression remained unchanged

following SMP treatment (Fig 4C). In contrast, these markers were clearly expressed in M2 macrophages (p<0.0001), with a notably higher expression of CD209 (Fig 4D and S2C Fig in S1 File).

Overall, these results indicate no polarizing effect of the SMPs, and the basal expression of these surface markers is unaffected. This finding suggests that, at least at the 1:3 cell-to-particle ratio used, SMPs do not pose a pro-inflammatory risk. However, it is crucial to highlight that the expression of these markers on macrophages is influenced not only by the stimuli they encounter but also by the lineage of the cells [39]. This is particularly relevant for this cell line, whose phenotypic and functional heterogeneity has been noted since its initial characterization [40]. This study is the first report evaluating this panel of surface markers in MH-S cells, which provides information mainly to learn more about the cell line.

## The stimulation with SMPs regulates the production of nitric oxide in alveolar macrophages previously polarized to M1

Next, it was important to investigate the effects of SMPs on macrophages regarding their production of NO and cytokines, glucose consumption, and lactate release, key indicators of cellular activation. NO is an important product of activated macrophages in response to cytokines, microbial compounds, or both. In this study, we assessed the effects of SMPs on isolation, without microbial or adjuvant compounds. Due to the attributes given to glucans as biological response modifiers [41], we used soluble α-glucan (SS) and β-glucan (Zym) as stimulation controls. NO production remained relatively constant in M0 and M2 macrophages for the initial 24 hours, with a notable increase at 48 hours only (Fig 5A and 5C). However, introducing SMPs or other glucans did not significantly alter these NO levels at any time.

As anticipated, in macrophages previously polarized to M1 with LPS and INF-γ, the levels of NO augmented. For the M1 phenotype (Fig 5B), the NO levels were slightly higher and increased with time. Although the statistical analysis did not show a significant difference, a key observation was the reduction in NO levels at both the 24 and 48-hour time points following the addition of SMPs to M1 macrophages. This indicates that SMPs exert a modulatory effect in the context of prior pro-inflammatory stimulation. Similarly, neither the soluble α-glucan nor the β-glucan influenced the M1 phenotype.

These findings are consistent with prior research in the field. Niu et al [19], reported minimum levels of NO in RAW 264.7 macrophages stimulated with a purified-unmodified α-glucan from *Actinidia chinensis* roots. Additionally, Kim et al. [42] demonstrated that opsonized zymosan particles alone do not induce significant NO production in macrophages unless combined with stimuli such as LPS and IFN-γ. This underscores the complexity of macrophage activation and the specific roles that different stimuli play in modulating their responses.

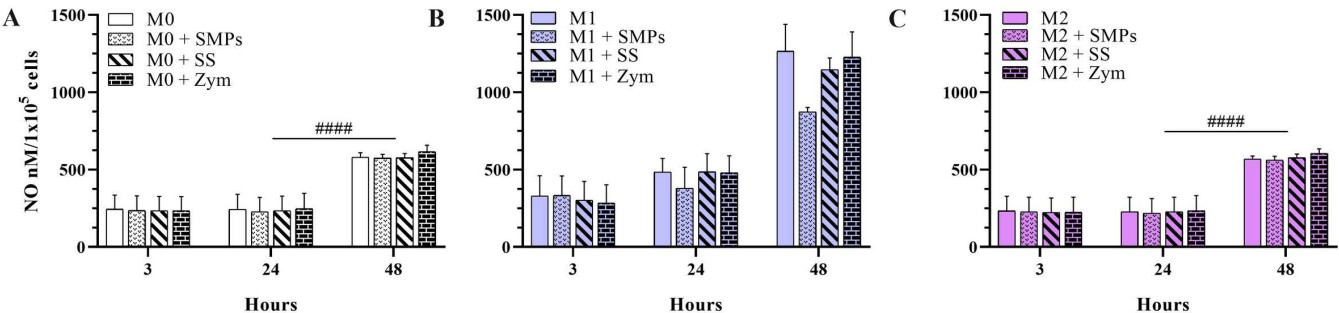

**Fig 5. Production of nitric oxide in alveolar macrophages of MH-S cell line stimulated with SMPs and glucans evaluated by Griess. A)** M0 MH-S alveolar macrophages. **B)** M1 MH-S alveolar macrophages. **C)** M2 MH-S alveolar macrophages. Smooth bars indicate the controls for each phenotype; dotted bars indicate macrophages stimulated with SMPs 1:3 ratio; striped bars indicate macrophages stimulated with soluble starch (10 µg/ml); bars with bricks indicate macrophages stimulated with zymosan (10 µg/ml). Bars indicate means±SEM from two independent experiments (n=6). Two-way ANOVA and Tukey's multiple comparisons (#### p<0.0001).

## The addition of SMPs decreases glucose consumption and lactate release in alveolar macrophages previously polarized to M1

Estimating glucose consumption and lactate release in this cell line was interesting because these metabolic activities are known to be associated with macrophage phenotype and function [43]. Our observations regarding NO production revealed that the most significant effects of SMPs were on M1 macrophages (Fig 6A). The M1 macrophages had less glucose consumption and lactate release, contrary to the expected for this phenotype, which typically relies on glycolysis for energy production, especially when activated.

However, it is essential to consider that the alveolar macrophages naturally reside in a glucose-poor environment, making them less capable of utilizing glycolysis. Instead, they depend more on other substrates to fuel oxidative phosphorylation (OXPHOS) for their energy supply [44]. Interestingly, the M1 group stimulated with the SMPs showed a significantly lower glucose consumption at 3 h (0.6%, $p < 0.05$) (Fig 6A). This group also released substantially lower amounts of lactate after adding SMPs ($p < 0.0001$). This behavior was similarly observed in M1 macrophages at 48 hours ($p < 0.01$), as well as in M0 and M2 macrophages (Fig 6A and 6B). Previous reports have suggested that reduced glucose consumption and lactate release indicate that alveolar macrophages are not utilizing the glycolytic pathway [44]. The decrease in glucose consumption and lactate release we observed at 3 h in the SMP-treated group—especially in M1 macrophages—suggests that these macrophages may use SMPs as an alternative carbon source for OXPHOS.

## The addition of SMPs modulates early cytokine secretion in MH-S alveolar macrophages

To provide a comprehensive overview of the effects of SMPs on M0, M1, and M2 alveolar macrophages, we examined the production of cytokines at 3, 24, and 48 hours. Fig 7A–7C show the pro-inflammatory cytokines, while Fig 7D–7E illustrate the anti-inflammatory cytokines. Consistent with our previous findings regarding the impact of SMPs on NO production,

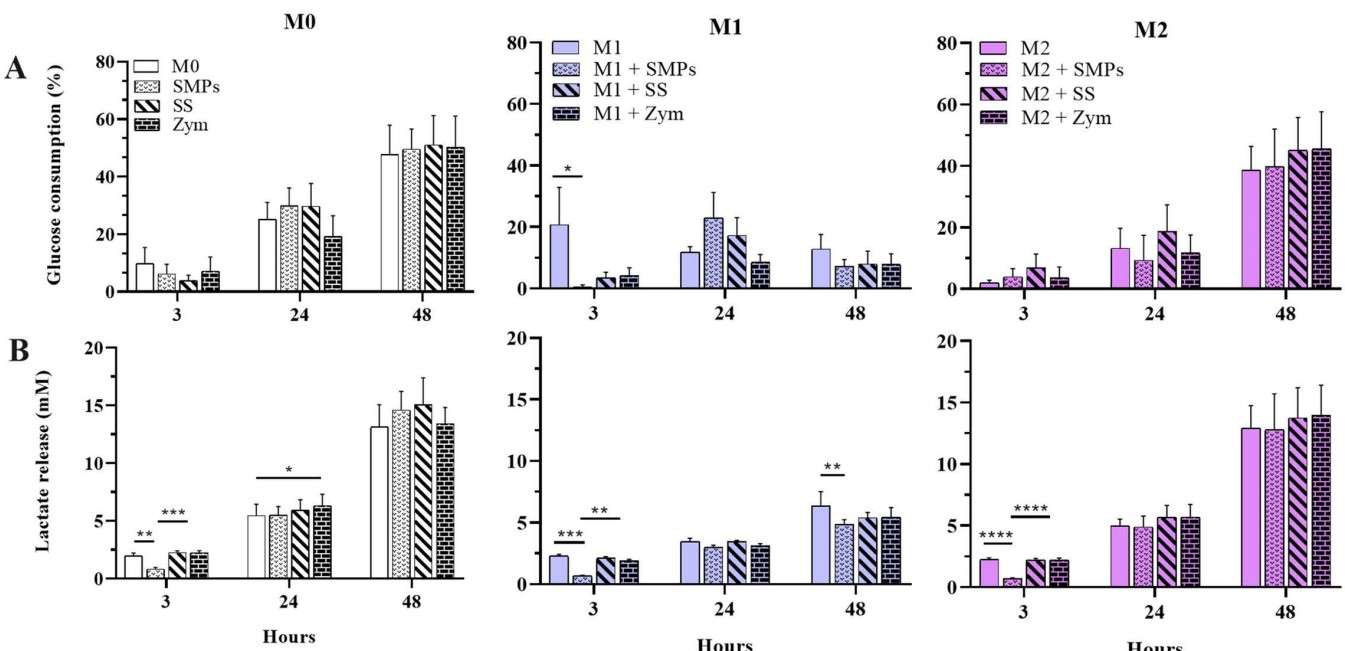

**Fig 6. Glucose consumption and lactate release of alveolar macrophages of MH-S cell line stimulated with SMPs and other glucans. A)** Glucose consumption. **B)** Lactate release. Bars indicate means ± SEM of two independent experiments (n = 6). Two-way ANOVA and Tukey's multiple comparisons (*$p < 0.05$, **$p < 0.01$, ***$p < 0.001$, ****$p < 0.0001$).

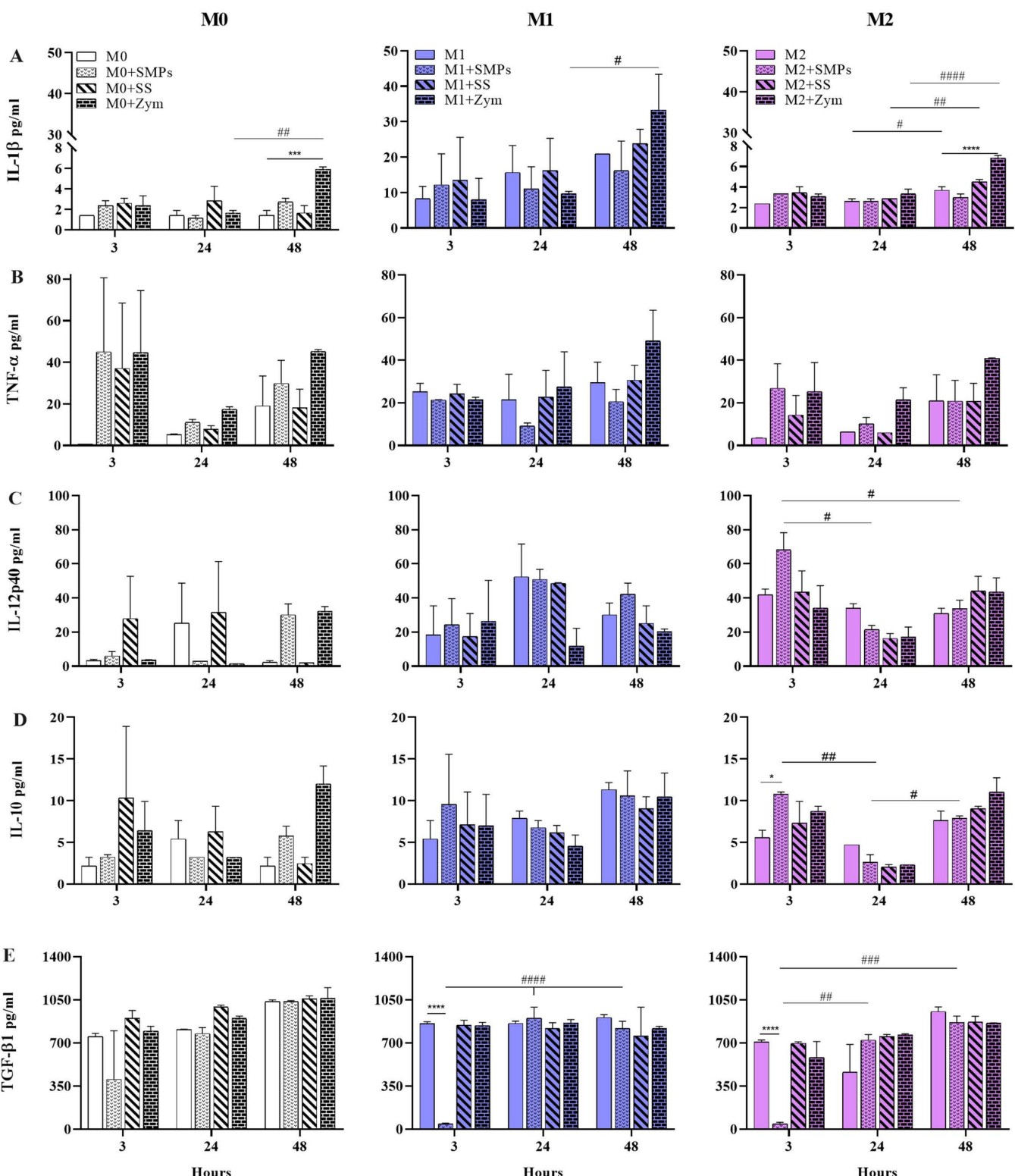

**Fig 7. Secretion of cytokines in alveolar macrophages of MH-S cell line stimulated with SMPs and other glucans. A-C)** Secretion of proinflammatory cytokines **D-E)** Secretion of anti-inflammatory cytokines. Smooth bars indicate the controls for each phenotype; dotted bars indicate macrophages stimulated with SMPs 1:3 ratio; striped bars indicate macrophages stimulated with soluble starch (10 μg/ml); bars with bricks indicate

macrophages stimulated with zymosan (10 µg/ml). Bars indicate means ± SEM from two independent experiments (n = 6). Two-way ANOVA and Dunnett´s multiple comparisons (differences between the control group and the treatments *p < 0.05, ***p < 0.001, ****p < 0.0001). Two-way ANOVA and Tukey's multiple comparisons (differences among times #p < 0.05, ##p < 0.01, ###p < 0.001, ####p < 0.0001).

glucose consumption, and lactate release in macrophages polarized to an M1 phenotype, we observed a decrease in the production of IL-1β and TNF-α at 24 and 48 hours upon the addition of SMPs. However, this change was not statistically significant (Fig 7A and 7B). After 48 h of stimulation with Zym, we observed the secretion of IL-1β and TNF-α in all phenotypes, which coincides with previous reports about the mechanisms of this compound to induce inflammation [45]. In contrast, M0 and M2 macrophages exhibited increased TNF-α secretion after 3 hours of stimulation with SMPs (Fig 7B). These findings are significant for our research on SMPs in the context of tuberculosis infection [27,46], as this response is crucial for the early immune defense against intracellular agents like *Mycobacterium tuberculosis* (Mtb) in susceptible cell lineages [47].

Regarding IL-12p40, it has been documented that its synthesis by macrophages is rapidly triggered (within 3–6 hours) after stimulation with LPS and IFN-γ, compared to macrophages in the M0 state [48], which aligns with our results (Fig 7C). In M2 macrophages, the addition of SMPs at the 3-hour mark increased the production of this cytokine. Compared to the IL-12p70 heterodimer, IL-12p40 has immunomodulatory effects on both pro-inflammatory and anti-inflammatory responses in macrophages and a chemoattractant role for other macrophages [49]. Importantly, IL-12p40 has been shown to play a critical role in activating IFN-γ secreting CD4 + T cells, which, in turn, activate macrophages to enhance their mycobactericidal capabilities [50].

In our analysis of anti-inflammatory cytokines (Fig 7D–7E), we found that the addition of SMPs led to an increase in IL-10 levels at 3 hours in M1 macrophages, and this increase was significant in M2 macrophages (p < 0.05). This finding is critical because IL-10 is essential for developing tolerance to inhaled antigens and maintaining lung homeostasis, which helps prevent excessive inflammation and tissue damage during infections [51].

Additionally, we examined the secretion of TGF-β1, a cytokine associated with anti-inflammatory responses and tissue repair. It is also linked to fibroblast differentiation, increased collagen secretion, and fibrosis [52]. We found that the addition of SMPs had no impact on TGF-β1 levels in M0 macrophages. However, at 3 hours, TGF-β1 levels were significantly reduced in M1 and M2 macrophages. This effect seems to be reversible since this significant decrease is only observed at 3 h (Fig 7E), while at 24 and 48 h, TGF-β1 levels are like the other groups, which is important since the complete elimination of this cytokine may affect pulmonary homeostasis [53]. The reduction of TGF-β1 at early time points in M1 and M2 macrophages upon the addition of SMPs (p < 0.0001) indicates the modulatory properties of this cytokine and suggests the potential benefits of using SMPs as immunomodulators in contexts involving intracellular infections, supported by studies demonstrating that a decrease in TGF-β1 enhances bacterial clearance [54].

Since we utilized unmodified SMPs without incorporating other molecules or infectious agents, these findings highlight the significance of SMPs as immunomodulators in models of intracellular infections. This is particularly relevant to our work, as we have used these SMPs as antigen carriers, boosters, and BCG adjuvant in a murine model of tuberculosis.

## Conclusions and outlook

SMPs are safe for alveolar macrophages of the MH-S cell line, and the process by which they are internalized is phagocytosis. All the findings indicate that these alpha-glucan particles have a modulatory effect on macrophages previously polarized to a proinflammatory phenotype (M1) being capable of regulating the secretion of proinflammatory cytokines and nitric oxide and reducing glucose consumption and lactate release, which is associated with anti-inflammatory activity. In M0 and M2 macrophages, the addition of the SMPs also had a regulatory effect, inducing a mixed secretion of cytokines such as TNF-α, IL-10 and IL-12p40, as well as a significant decrease of TGF-β1, which opens the prospects for using

these SMPs in prophylactic or therapeutic vaccines directed to control intracellular infections in the lung. This is of utmost importance since the induction of immune regulation is sought with the use of polysaccharides without compromising efficacy and safety.

Importantly, we know the limitations that this *in vitro* study with a cell line has as an environment far from the natural environment of alveolar macrophages in the mouse or human lung, where cells experience three-dimensional contact with proteins and epithelial cells, biomechanical forces, as well as dynamic nutrient and waste gradients. Each of these factors can influence how the cells behave. However, *in vitro* studies represent the best approach for functional screening of molecules and drugs in preclinical research, allowing for control of growth, maintenance, observation and manipulation of cells which provides relevant information about events that occurred after the addition of molecules, and consequently a comprehensive interpretation of *in vivo* events too. A comprehensive understanding and visualization of the effects of alpha-glucan particles on the phenotype and activation of alveolar macrophages are essential for elucidating their *in vivo* mechanisms, effects, and potential applications as carriers or vaccine adjuvants for nasal or pulmonary administration.

## Supporting information

**S1 File. Supplementary figures.**
(DOCX)

**S2 File. Raw data underlying figures 1–7.**
(XLSX)

## Acknowledgments

We want to thank to Dr. Iris S Paredes-González for technical advice in glucose consumption and release lactate assays, to C. Dr. J Carlos Blanco-Camarillo for advice on flow cytometry analysis, to Juan Carlos León and Dr. Rodolfo Paredes for technical support for MET, and Dr. Daniel Guillén for technical assistance.

## Author contributions

**Conceptualization:** Alejandra Barrera-Rosales, Mayra Silva-Miranda, Edgar Zenteno, Sergio Sánchez, Rogelio Hernández-Pando, Romina Rodríguez-Sanoja, Silvia Moreno-Mendieta.

**Formal analysis:** Alejandra Barrera-Rosales, Silvia Moreno-Mendieta.

**Funding acquisition:** Romina Rodríguez-Sanoja, Silvia Moreno-Mendieta.

**Investigation:** Alejandra Barrera-Rosales, Silvia Moreno-Mendieta.

**Methodology:** Alejandra Barrera-Rosales, Dulce Mata-Espinosa, Vanessa Villegas-Ruiz.

**Project administration:** Silvia Moreno-Mendieta.

**Resources:** Rogelio Hernández-Pando.

**Supervision:** Silvia Moreno-Mendieta.

**Validation:** Alejandra Barrera-Rosales, Romina Rodríguez-Sanoja, Silvia Moreno-Mendieta.

**Visualization:** Alejandra Barrera-Rosales, Silvia Moreno-Mendieta.

**Writing – original draft:** Alejandra Barrera-Rosales, Silvia Moreno-Mendieta.

**Writing – review & editing:** Alejandra Barrera-Rosales, Dulce Mata-Espinosa, Vanessa Villegas-Ruiz, Mayra Silva-Miranda, Edgar Zenteno, Sergio Sánchez, Rogelio Hernández-Pando, Romina Rodríguez-Sanoja, Silvia Moreno-Mendieta.

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
