## [Decision Letter · Decision Letter 0]

PONE-D-25-09559Unveiling the immunomodulatory properties of starch microparticles on alveolar macrophagesPLOS ONE

Dear Dr. Moreno-Mendieta,

Thank you for submitting your manuscript to PLOS ONE. After careful consideration, we feel that it has merit but does not fully meet PLOS ONE’s publication criteria as it currently stands. Therefore, we invite you to submit a revised version of the manuscript that addresses the points raised during the review process.

We look forward to receiving your revised manuscript.

Kind regards,

Shengwei Sun, Ph.D.

Academic Editor

PLOS ONE

**Journal Requirements:**

1. When submitting your revision, we need you to address these additional requirements. Please ensure that your manuscript meets PLOS ONE's style requirements, including those for file naming. The PLOS ONE style templates can be found at https://journals.plos.org/plosone/s/file?id=wjVg/PLOSOne_formatting_sample_main_body.pdf and https://journals.plos.org/plosone/s/file?id=ba62/PLOSOne_formatting_sample_title_authors_affiliations.pdf 2. Thank you for stating in your Funding Statement: SMM was supported by grant A1-S-14446 Ciencia Básica (SEP-CONAHCyT), RRS was supported by grants A1-S-9849 Ciencia Básica (SEP-CONAHCyT) and IN 216722 PAPIIIT/DGAPA/UNAM. ABR was supported by doctoral scholarship 894774 (CONAHCyT). Please provide an amended statement that declares *all* the funding or sources of support (whether external or internal to your organization) received during this study, as detailed online in our guide for authors at http://journals.plos.org/plosone/s/submit-now.  Please also include the statement “There was no additional external funding received for this study.” in your updated Funding Statement. Please include your amended Funding Statement within your cover letter. We will change the online submission form on your behalf. 3. When completing the data availability statement of the submission form, you indicated that you will make your data available on acceptance. We strongly recommend all authors decide on a data sharing plan before acceptance, as the process can be lengthy and hold up publication timelines. Please note that, though access restrictions are acceptable now, your entire data will need to be made freely accessible if your manuscript is accepted for publication. This policy applies to all data except where public deposition would breach compliance with the protocol approved by your research ethics board. If you are unable to adhere to our open data policy, please kindly revise your statement to explain your reasoning and we will seek the editor's input on an exemption. Please be assured that, once you have provided your new statement, the assessment of your exemption will not hold up the peer review process. 4. We notice that your supplementary figures are uploaded with the file type 'Figure'. Please amend the file type to 'Supporting Information'. Please ensure that each Supporting Information file has a legend listed in the manuscript after the references list.

Reviewers' comments:

Reviewer's Responses to Questions

**Comments to the Author**

1. Is the manuscript technically sound, and do the data support the conclusions?

Reviewer #1: Yes

2. Has the statistical analysis been performed appropriately and rigorously? 

Reviewer #1: Yes

3. Have the authors made all data underlying the findings in their manuscript fully available?

Reviewer #1: Yes

4. Is the manuscript presented in an intelligible fashion and written in standard English?

Reviewer #1: Yes

5. Review Comments to the Author

**Reviewer #1:**  I have no Competing Interests.

{1.} following sentence { lines 120-121} is not clear. Authors can check and correct it.

the crystals of formazan were dissolved with SDS and the lecture was made at 570 nm.

6. PLOS authors have the option to publish the peer review history of their article (what does this mean? ). If published, this will include your full peer review and any attached files.

**Do you want your identity to be public for this peer review?** For information about this choice, including consent withdrawal, please see our Privacy Policy .

Reviewer #1: No

---

## [Author Response · Author response to Decision Letter 1]

6 Jun 2025

Journal Requirements:

After revising the PLOS ONE style templates, we made the corresponding changes to meet requirements. The changes appear in the 'Revised Manuscript with Track Changes' file and are applied in the “Manuscript” file.

According to the guide for authors in the Financial Disclosure Statement section, all the funding or sources of support received during this study are entered in the submission form section and were eliminated from the manuscript file.

The new amended statement includes specific grant numbers, initials of authors who received each award, full names of entities that funded the study or authors, and the statement: “There was no additional external funding received for this study.” As follows:

A1-S-14446, SMM, Ciencia Básica Fondo Sectorial de la Secretaría de Educación Pública y el Consejo Nacional de Humanidades, Ciencias y Tecnologías (SEP-CONAHCyT)

A1-S-9849, RRS, Ciencia Básica Fondo Sectorial de la Secretaría de Educación Pública y el Consejo Nacional de Humanidades, Ciencias y Tecnologías (SEP-CONAHCyT)

IN 216722, RRS, Programa de Apoyo a Proyectos de Investigación e Innovación Tecnológica/Dirección General Asuntos del Personal/Universidad Nacional Autónoma de México (PAPIIIT/DGAPA/UNAM)

894774, ABR, Doctoral Scholarship Consejo Nacional de Humanidades, Ciencias y Tecnologías (CONAHCyT)

There was no additional external funding received for this study.

As previously declared, yes, all data are fully available without restriction and the statement is completed in the submission form. All data underlying the reported findings are provided as part of the submitted manuscript or supporting information files. Our experimental data do not need to be deposited in public-specific repositories because they are not microarray data or gene sequences.

4. We notice that your supplementary figures are uploaded with the file type 'Figure'. Please amend the file type to 'Supporting Information'. Please ensure that each Supporting Information file has a legend listed in the manuscript after the references list.

The amended file was uploaded as supporting information, and the legends were listed in the manuscript after the references list.

The reference list was carefully revised and edited to ensure that it is complete and correct. No retracted papers have been cited. Editions appear in the 'Revised Manuscript with Track Changes' file.

Reviewers' comments:

Reviewer #1:

{1.} following sentence { lines 120-121} is not clear. Authors can check and correct it.

the crystals of formazan were dissolved with SDS and the lecture was made at 570 nm.

The sentence was edited and appears in lines 124 and 125 of the “Manuscript” file and is indicated in lines 129 and 130 of the 'Revised Manuscript with Track Changes' file.

All figures were processed with PACE before submission to meet requirements.

---

## [Editor Report · Decision Letter 1]

Unveiling the immunomodulatory properties of starch microparticles on alveolar macrophages

PONE-D-25-09559R1

Dear Dr. Moreno-Mendieta,

We’re pleased to inform you that your manuscript has been judged scientifically suitable for publication and will be formally accepted for publication once it meets all outstanding technical requirements.

Kind regards,

Shengwei Sun, Ph.D.

Academic Editor

PLOS ONE
---

## [Editor Report · Acceptance letter]

PONE-D-25-09559R1

PLOS ONE

Dear Dr. Moreno-Mendieta,

I'm pleased to inform you that your manuscript has been deemed suitable for publication in PLOS ONE. Congratulations! Your manuscript is now being handed over to our production team.

Kind regards,

on behalf of

Dr. Shengwei Sun

Academic Editor

PLOS ONE